# Proportions of Polyunsaturated Fatty Acids in Umbilical Cord Blood at Birth Are Related to Atopic Eczema Development in the First Year of Life

**DOI:** 10.3390/nu13113779

**Published:** 2021-10-25

**Authors:** Malin Barman, Mia Stråvik, Karin Broberg, Anna Sandin, Agnes E. Wold, Ann-Sofie Sandberg

**Affiliations:** 1Food and Nutrition Science, Department of Biology and Biological Engineering, Chalmers University of Technology, 41296 Gothenburg, Sweden; mia.stravik@chalmers.se (M.S.); ann-sofie.sandberg@chalmers.se (A.-S.S.); 2Institute of Environmental Medicine, Karolinska Institutet, 17177 Stockholm, Sweden; karin.broberg@ki.se; 3Occupational and Environmental Medicine, Department of Laboratory Medicine, Lund University, 22363 Lund, Sweden; 4Pediatrics, Department of Clinical Sciences, Umeå University, 90187 Umeå, Sweden; anna.sandin@umu.se; 5Department of Infectious Diseases, Institute of Biomedicine, The Sahlgrenska Academy, University of Gothenburg, 40530 Gothenburg, Sweden; agnes.wold@microbio.gu.se

**Keywords:** diet, pregnancy, cord blood, fatty acids, n-3 PUFAs, n-6 PUFAs, arachidonic acid, phospholipids, atopic eczema, NICE birth cohort

## Abstract

Atopic eczema, the most common atopic disease in infants, may pave the way for sensitization and allergy later in childhood. Fatty acids have immune-regulating properties and may regulate skin permeability. Here we examine whether the proportions of fatty acids among the infant and maternal plasma phospholipids at birth were associated with maternal dietary intake during pregnancy and development of atopic eczema during the first year of age in the Nutritional impact on Immunological maturation during Childhood in relation to the Environment (NICE) birth cohort. Dietary data were collected with a semi-quantitative food frequency questionnaire, fatty acids were measured with GC-MS and atopic eczema was diagnosed by a pediatric allergologist at 12 months of age. We found that higher proportions of n-6 PUFAs (including arachidonic acid) but lower proportions of n-3 PUFAs (including DPA) in the infant’s phospholipids at birth were associated with an increased risk of atopic eczema at 12 months of age. The n-6 and n-3 PUFAs were related to maternal intake of meat and fish, respectively. Our results suggest that prenatal exposure to unsaturated fatty acids is associated with eczema development in the infant. Maternal diet during pregnancy may partly explain the fatty acid profiles *in utero*.

## 1. Introduction

Atopic eczema is the most common skin disease in children, affecting up to 20% of all children and up to 3% of adults [1]. Atopic eczema is a multifaceted, chronic, inflammatory skin condition with age-specific distribution patterns [1]. The first manifestations of atopic eczema usually appear early in life and often precede other allergic manifestations such as allergic rhinitis and asthma [2]. The etiology of atopic eczema is not completely understood but depends on complex interactions between genetic and different environmental and lifestyle factors. Filaggrin is a protein essential for healthy skin barrier function. Loss-of-function mutations in the filaggrin (*FLG*) gene have been associated with higher total IgE levels, sensitization to more allergens, and increased risk of atopic diseases [3]. This suggests that skin barrier function is one important factor in atopic eczema development. 

One environmental factor that has been implicated in allergic diseases is diet [4]. Since atopic eczema often appears during the first year of life, maternal diet during pregnancy has been suggested to be of importance for the development of this disease in the infant [5,6]. Food contains certain nutrients that have immunomodulatory properties, such as different types of fatty acids. Fatty acids may be saturated, monounsaturated (MUFA), or polyunsaturated (PUFA). The PUFAs belong to either the n-6 or n-3 series. Different n-6 PUFAs may be endogenously synthesized by elongation of the precursor linoleic acid (LA, 18:2 n-6), which is a fatty acid that is abundant in vegetable oils and margarine. The n-3 PUFAs, on the other hand, may be synthesized from the n-3 precursor α-linolenic acid (ALA, 18:3 n-3), which is found in, for example, rapeseed oil. This elongation process is more effective in pregnant women, possibly due to the increased need for long-chain n-3 PUFAs during infancy [7,8]. In addition to the fatty acid elongation process that occurs in the human body, long-chain n-3 PUFAs are derived from intake of fish and other seafood. 

Both n-6 and n-3 PUFAs have well-known immunomodulatory effects [5,6,9,10]. For example, PUFAs bind to fatty acid receptors (such as the ligand-activated transcription factor peroxisome proliferator-activated receptors, PPARs) [11,12] on the membranes of immune cells [13,14]. PUFAs also act as precursors for different lipid mediators, such as the eicosanoids. While eicosanoids produced from the n-6 PUFA arachidonic acid promote allergic sensitization and allergic inflammation, it has been suggested that the n-3 PUFAs act to oppose these actions [6]. Based on the molecular and cellular mechanisms of the eicosanoids produced from arachidonic acid, Black and Sharp, in 1997, already suggested a causal linkage between increased n-6 PUFA intake and increased prevalence of allergic diseases [4]. In a review published in 2017, Miles and Calder further discussed the association between early n-3 PUFA exposure and allergic disease risk and suggested that eating oily fish or fish oil supplements during pregnancy could be a strategy to prevent infant and childhood allergic diseases [6]. This hypothesis is supported by results from observational studies that have reported protective effects of maternal fish intake during pregnancy on atopic eczema development in infants [15,16]. 

Here, the aims were to: (1) examine the associations between the fatty acid profiles of maternal and infant cord plasma at delivery and the incidence of atopic eczema during the first year of life in a large Swedish birth cohort; and (2) examine the correlations between maternal and infant plasma phospholipid fatty acid proportions, as well as the association between these proportions and maternal dietary intake during pregnancy.

## 2. Materials and Methods

### 2.1. Study Population

The current study is based on the Nutritional impact on Immunological maturation during Childhood in relation to the Environment (NICE) birth cohort study conducted in Luleå, northern Sweden [17]. Pregnant women who were able to communicate in Swedish and planned on giving birth at Sunderby Hospital were informed about the birth cohort during their first visit to the local maternity clinic in GW 10–12. Recruitment was conducted between February 2015 and March 2018, at the hospital in connection with a routine ultrasound at GW 18. In total, 655 pregnant women were included in the NICE birth cohort. Three women gave birth to twins, five fetuses died before birth, one woman had a late miscarriage after inclusion in the birth cohort, and one woman withdrew her participation. Thus, 645 singleton liveborn infants were included in the birth cohort. Of these, 16 families participated in the birth cohort with two subsequent pregnancies and were, thus, excluded from the current study. Plasma samples from the infants’ umbilical cords were available for fatty acid analyses for 290 of the 629 families eligible for inclusion in the current study, and 249 of these attended the 12-month follow-up and could be classified either as having atopic eczema or being non-allergic. Infants who did not have atopic eczema but had other allergic manifestations (such as food allergy or asthma) or were sensitized (N = 43) were excluded from the statistical analyses because they could not be classified as non-allergic. This resulted in a total of 206 infants being included in the statistical calculations in the current study, of which 14 were diagnosed with atopic eczema (see Figure 1 for flow chart). Plasma samples from the mothers were available for 193 of the mother and infant pairs, i.e., 13 in the allergic group and 180 in the non-allergic control group.

### 2.2. Collection of Maternal and Infant Blood

Infant cord blood was collected from the umbilical cord directly after delivery. The cord was clamped and severed before cord blood was squeezed out into 6 mL EDTA tubes (Becton Dickinson, Franklin Lakes, NJ, USA). Peripheral venous blood samples from the mothers were collected in 10 mL EDTA vacutainer tubes (Becton Dickinson) in connection with the delivery (usually when arriving at the delivery ward). Blood samples were stored refrigerated (4 °C) at the delivery ward until transported to the research laboratory, where they were centrifuged and the plasma was decanted, aliquoted, and stored at −80 °C. The research laboratory was only staffed on weekdays, which meant that samples acquired during deliveries that occurred during weekends could be stored at 4 °C for up to 3 days before centrifugation and subsequent freezing.

### 2.3. Analysis of Fatty Acid Proportions in Plasma Phospholipids

Plasma samples (200 µL) were thawed slowly in cold water, vortexed, and mixed with 50 µL of internal standard (fatty acid 17:0, 1 mg/mL). Fatty acids were extracted by adding 4 mL of chloroform:methanol 1:1 (*v*/*v*) and 2 mL of 0.5% NaCl. After vortexing and centrifugation, 1 mL of the chloroform phase was collected and evaporated at 40 °C under nitrogen gas. The remaining precipitate was dissolved in 200 μL chloroform and run through solid-phase extraction (SPE) columns (Supelco, Bellefonte, PA, USA, NH_2_, 500 mg/3 mL), preconditioned with 2 × 2 mL hexane. The columns were rinsed with 2 × 2 mL chloroform:isopropanol 2:1 (*v*/*v*) and 2 × 2 mL of 2% acetic acid in diethyl ether. Thereafter, the adsorbed phospholipids were eluted with 2 × 2 mL of methanol. Methylation of fatty acids was carried out as follows: the methanol was evaporated and the precipitate was dissolved in 2 mL of toluene and 2 mL of 10% acetyl chloride in methanol and vortexed for 1 min. The mixture was incubated at 70 °C for 2 h with vortexing every 30 min. Thereafter, 1 mL of Milli-Q water and 2 mL of petroleum ether were added to the samples. After vortexing and centrifugation, 3 mL of the organic phase was collected and evaporated at 40 °C under nitrogen gas.

### 2.4. Analysis of Fatty Acids by GC-MS

The fatty acids were separated by gas chromatography-mass spectrometry (GC-MS) using the 7890A GC-system, 5975C inert XL EI/CI MSD with Triple-Axis Detector, and 7693 Autosampler, together with a VF-WAXms GC column (30 m × 0.25 mm × 0.25 µm; P/N 7FD-G033-05) (all from Agilent Technologies Inc., Santa Clara, CA, USA). The liner used was the Ultra Inert Inlet Liner, Low PSI drop, wool (P/N 5190-3165; Agilent Technologies). The Non-Stick Bleed/Temperature-Optimized Non-Stick 11-mm Septa was used (Agilent Technologies). 

Samples were dissolved in 100 μL iso-octane for the GC analysis. Three quality control samples were analyzed in parallel in each run. The standard GLC-463 mixed fatty acid methyl esters (Nu-Chek Prep, Elysian, MN, USA) were dissolved in toluene and used as an external standard for peak evaluation. The GC oven program consisted of: initial temperature of 100 °C, ramping at 4 °C/min to 205 °C and then at 1 °C/min to 230 °C with a 5-min hold. The instrumental settings were: helium as the carrier gas; inlet heater at 275 °C; pressure, 10.523 psi; total flow, 14 mL/min, septum purge flow, 3 mL/min; split flow, 10 mL/min; and split ratio, 10. The injection volume was 1 µL per sample, with an oven run time of 56 min. 

The fatty acids in the phospholipids were identified by comparing the retention times of the external standard with the retention times of the samples, as well as a mass spectrum analysis of each peak and comparison with a library of the mass spectra of known fatty acids, which is maintained in the laboratory. 

The following fatty acids were identified and quantified in the phospholipid fraction of the plasma: 14:0, 20:0, 18:1 n-9, 18:1 n-7, 18:2 n-6 (LA), 18:3 n-3 (ALA), 20:3 n-6 (DGLA), 20:4 n-6 (AA), 20:5 n-3 (EPA), 22:0, 22:4 n-6, 22:5 n-6, 22:5 n-3 (DPA) and 22:6 n-3 (DHA). The concentration of each fatty acid was calculated using the concentration of the internal standard 17:0. The proportions of specific fatty acids were expressed as the concentration of the particular fatty acid relative to the concentration of all 14 fatty acids. Fatty acids 16:0 and 18:0 were not quantified due to contamination from the SPE columns and, therefore, are not included in the total sum of fatty acids.

### 2.5. Dietary Assessments

Maternal daily intake of food and nutrients during pregnancy was assessed using a web-based, semi-quantitative, food frequency questionnaire (Meal-Q), as previously described in detail [18,19]. The questionnaire, which was distributed to the subjects when they were around GW 34, sought information regarding food intake during the previous month, measured on a frequency scale that ranged from no intake/intake less than once per month to ≥5 times/day. The intake of food in grams per day was calculated using either reported intake quantities or, if no quantities were specified, standard portions from the Swedish Food Composition Database [20].

### 2.6. Clinical Examination

Atopic eczema during the first year of life was diagnosed by an experienced pediatric allergologist (author AS) at a clinical visit that was scheduled at 12 months of age. Diagnosis followed the criteria proposed by Williams and coworkers [21,22,23]. The major criterion was an itchy condition or parental report of scratching/rubbing by the child. This resulted in a diagnosis when present in combination with three or more of the following criteria: (1) history of infant eczema that was spreading generally or found in typical areas and creases, such as the cheeks, folds of the elbows, rear of knees, front of ankles or around the neck; (2) history of asthma or other atopic disease or a history of atopic disease in a first-degree relative; (3) history of generally dry skin in the previous year; and (4) visible infant eczema that was generally spreading or located in typical areas and creases, such as the cheeks, folds of the elbows, behind the knees, front of ankles or around the neck.

### 2.7. Assessment of Filaggrin Gene (FLG) Mutations

DNA extraction was performed on infant EDTA-treated blood or blood cells using the Omega E.Z.N.A. Blood DNA Mini Kit (Omega Bio-tek Inc., Norcross, GA, USA). The 260/280 nm and the 260/230 nm absorbance ratios were measured with the Nanodrop 1000 Spectrophotometer (Thermo Fisher Scientific, Waltham, MA, USA). 

The *FLG* null mutations R501X, R2477X, S3247X, and 2282del4 (the most common mutations in Europe) were analyzed [24]. For R501X, R2477X, and S3247X the qPCR was carried out in a total reaction volume of 5 µL, containing: 0.125 µL of 40× custom-made SNP genotyping assay (Thermo Fisher Scientific), 2.5 µL of 2× TaqMan Master Mix (Thermo Fisher Scientific), and 2 µL of DNA (5 ng/µL). The PCR program was as follows: 10 min at 95 °C, followed by 45 cycles of 15 s at 92 °C, and a final step of 1 min and 30 s at 60 °C. 

For 2282del4, 0.125 µL of the SNP assay, 2.5 µL of TaqMan MasterMix, 0.25 µL of a 10-µM stock of second primer (for increased specificity of the reaction), and 2 µL of DNA (5 ng/µL) were added to the reaction mixture. The PCR program was as above, except that the final step was for 1 min at 60 °C. The programs were run on the ABI 7900 instrument. The primer pairs for the different assays are listed in Appendix A. As a positive control, a synthesized DNA construct that included the R501X, R2447X, S3247X, and 2282del4 rare alleles and surrounding sequences was included in each qPCR run in duplicate (GeneArt; Thermo Fisher Scientific).

### 2.8. Data Variables

Data on maternal and birth characteristics were extracted from electronic hospital records. All data were anonymized before processing. Allergic heredity and pet ownership were assessed as part of a structured interview conducted during the study visit at 12 months of age. Data on the number of siblings in the house and residential area were extracted from questionnaires filled in by both parents around GW 20–25.

### 2.9. Statistical Methods

All statistical analyses were performed using the IBM SPSS Statistics ver. 27.0 (IBM, New York, NY, USA), the R ver. 3.6.2 software (R Foundation for Statistical Computing, Vienna, Austria), and Umetrics SIMCA ver. 16.0.1 (Sartorius Stedim Data Analytics AB, Umeå, Sweden). Participants’ characteristics were analyzed using the χ^2^ test or Fisher’s exact test for categorical variables and the Mann–Whitney *U*-test for continuous variables. Differences in the levels of plasma fatty acids were analyzed using the Mann–Whitney *U*-test. Correlations were calculated by Spearman’s rho. Logistic regression models were used to analyze associations between proportions of fatty acids and atopic eczema. One unadjusted model was performed and two adjusted models, one with adjustment for any allergy within the family (parent or sibling) and one with adjustment for *FLG* loss-of-function mutation. The logistic regression models were performed on standardized fatty acid levels (per interquartile range, IQR). Therefore, the results refer to a change in an interquartile range for each fatty acid proportion. Partial least square (PLS) was used to investigate which of the food variables were most strongly related to the infant and maternal plasma proportions of n-3 and n-6 PUFAs. Orthogonal partial least square with discriminant analysis (OPLS-DA) was used to investigate which of the variables were most strongly related (positively or negatively) to atopic eczema.

## 3. Results

Cord blood and maternal peripheral venous blood were obtained at delivery from 206 infants and 196 mothers, respectively, and the fractions of different fatty acids in the plasma phospholipids were determined (expressed as the proportions of all measured fatty acids). Figure 2 and Appendix A show the proportions of various PUFAs in the infant and maternal plasma phospholipids.

As shown in Figure 2, compared to the maternal plasma phospholipids, the infant plasma phospholipids were enriched for the n-6 PUFAs arachidonic acid (AA, 20:4 n-6), dihomo-gamma-linolenic acid (DGLA, 20:3 n-6) and adrenic acid (22:4 n-6), as well as the n-3 PUFA docosahexaenoic acid (DHA, 22:6 n-3). In contrast, the maternal plasma phospholipids contained higher proportions of linoleic acid (LA, 18:2 n-6), α-linolenic acid (ALA, 18:3 n-3), eicosapentaenoic acid (EPA, 20:5 n-3), and docosapentaenoic acid (DPA, 22:5 n-3) (Figure 2, Appendix A).

Even though the proportions of many of the fatty acids differed substantially between the mothers and their infants (Figure 2), all the fatty acids, with the exception of α-linolenic acid, were strongly positively correlated (*p* < 0.001) between infants and mothers (Appendix A).

### 3.1. Prevalence of Eczema and Relationships between Eczema and Background Factors

The infants were clinically examined by a pediatric allergology specialist at 12 months of age. In total, 14 infants were diagnosed with atopic eczema, and 192 infants were classified as non-allergic and non-sensitized. Sensitized children without any allergy diagnosis and children with allergies other than eczema were excluded from the control group (N = 43). Children who had eczema and other allergic manifestations, such as food allergy (N = 5) or asthma (N = 3), were included in the eczema group. Thus, 8 of the 14 infants with eczema also had an additional atopic manifestation.

Table 1 lists the characteristics of the study participants, showing the differences between the allergic and non-allergic children. The vast majority of the infants were born by vaginal delivery and most were breastfed for at least 6 months. Children with eczema were more likely to have an allergic family member (*p* = 0.033). More specifically, allergy in a sibling was highly predictive of eczema development during the first year of life (*p* = 0.007). *FLG* loss of function mutation was not associated with the prevalence of atopic eczema. No other significant differences were found between the allergic and non-allergic children (Table 1).

### 3.2. Proportions of Fatty Acids in Plasma Phospholipids at Birth in Relation to Presence of Atopic Eczema during the First Year of Life

The fatty acid compositions of the plasma phospholipids in maternal venous plasma and infant cord plasma samples obtained at delivery were related to the diagnosis of atopic eczema at 12 months of age using OPLS. In the model, eczema was set as the outcome (*Y*) variable, and the proportions of individual fatty acids were set as explanatory (*X*) variables. Figure 3 shows a model that only includes those fatty acids with a variable of importance (VIP) value > 0.8; a model incorporating all fatty acids is shown in Appendix A. The variables that showed the strongest associations with atopic eczema were further investigated using univariate analysis. As shown in Figure 3, the sum of the n-6 PUFA proportions in the infant’s phospholipids was significantly positively associated with the risk of having atopic eczema at 12 months of age (*p* = 0.002). The same was true for the sum of the long-chain n-6 fatty acids (n-6 LCPUFA; *p* = 0.001), as well as in the case of arachidonic acid (*p* = 0.002). In contrast, higher proportions of the long-chain n-3 PUFA DPA in the infant’s cord serum correlated negatively with atopic eczema (*p* = 0.03). Other n-3 PUFAs also appeared at the opposite side of eczema in the plot, although they were not significantly associated with the non-allergic state. In general, maternal and infant fatty acids appeared on the same side of the plot, although the maternal fatty acid composition was more weakly associated with allergy in the infant than was the infant’s own fatty acid pattern (Figure 3). In addition, the presence of α-linolenic acid in the infant’s cord blood was significantly associated with a decreased risk of developing eczema (*p* = 0.02) (Appendix A), although this association was excluded by the VIP analysis due to a too-weak contribution to the model, such that it is not visible in Figure 3.

The proportions of fatty acids in the cord blood of infants who subsequently developed eczema or remained non-allergic are listed in Appendix A, while the corresponding values for the maternal plasma are shown in Appendix A. Neither monounsaturated fatty acids (MUFA) nor saturated fatty acids in the infant’s or mother’s blood at delivery were related to the risk of developing eczema during the first year of life (Appendix A).

### 3.3. Magnitude of the Association between Infant Cord Blood Fatty Acids and Atopic Eczema

The magnitude of the associations between the proportions of n-3 and n-6 fatty acids in the phospholipids present in infant cord serum at birth and atopic eczema were further explored with logistic regression models, without adjustment and with adjustment for either any allergic disease within the family (sibling or parent) or being a carrier of *FLG* loss-of-function mutation (Table 2). Primarily, arachidonic acid was found to account for the positive association between the infant’s proportions of n-6 PUFAs and long-chain n-6 PUFAs and eczema at 12 months of age. An increase of one IQR in the proportion of arachidonic acid in the infant cord plasma phospholipids corresponded to a 2.8-fold increase in the odds of having atopic eczema during the first year of life in the unadjusted model. After adjustment for allergy within the family or for *FLG* mutation, the odds were similar (OR = 2.6 in both models). Arachidonic acid was the dominant n-6 fatty acid in the infant cord blood phospholipids, representing 71% of the long-chain n-6 PUFAs and 56% of the n-6 PUFAs. Thus, an increase of one IQR in the proportion of long-chain n-6 PUFAs or total n-6 PUFAs resulted in 2.1-fold or 1.8-fold higher odds of developing atopic eczema, respectively in the unadjusted models with slightly reduced but similar results in the adjusted models. 

Conversely, an increase of one IQR in the proportion of ALA in the cord blood plasma phospholipids was associated with a 3.0-fold lower odds of having a diagnosis of atopic eczema at 12 months of age (2.9-fold after adjustment for allergic heredity while not significant after adjusting for *FLG*). An increase of one IQR in the proportion of DPA resulted in 2.6-fold lower odds, after adjusting for any allergy within the family, while not significant in the unadjusted model or in the model adjusted for *FLG* (Table 2). 

### 3.4. Associations between Fatty Acids in Maternal and Infant Cord Plasma and Maternal Food Intake

The proportion of fatty acids in plasma phospholipids is partially determined by the diet. For example, a diet that is high in fish gives a higher proportion of the long-chain fatty acids EPA, DHA, and DPA. Partial least squares (PLS) analysis was used to examine whether maternal intake of different fat-containing foods was associated with the proportions of PUFAs in the plasma phospholipids of the infants and mothers. The proportions of n-3 PUFAs or n-6 PUFAs were set as the outcome (Y) variables, while the mother’s reported intake of different fatty acid-rich foods was set as explanatory (X) variables (Figure 4). As is evident in Figure 4, the infant’s and mother’s proportions of n-6 PUFAs in their plasma phospholipids were positively associated with the maternal intake of different sorts of meat and chocolate. The total n-3 PUFAs in both the infant and maternal plasma were instead related to maternal intake of seafood, total fish, and fatty fish, which are known sources of n-3 PUFAs, as well as the intake levels of game meat, egg, cheese, and milk. Univariate statistical analysis using Spearman’s correlation showed the only significant correlations between maternal dietary food intake and infant proportions of fatty acids were a negative correlation between the infant proportions of n-6 PUFAs and maternal intake of fatty fish (Rho = −0.17, *p* = 0.02) and total fish (Rho = −0.16, *p* = 0.03). For maternal fatty acid proportions more correlations were significant in univariate statistics: the maternal proportions of n-3 PUFAs correlated positively with their intake levels of fatty fish (Rho = 0.22, *p* = 0.003), total fish (Rho = 0.21, *p* = 0.005), egg (Rho = 0.17, *p* = 0.02), and nuts and seeds (Rho = 0.19, *p* = 0.01), whereas they correlated negatively with the intake of chocolate (Rho = −0.17, *p* = 0.02). The maternal proportions of n-6 PUFAs, on the other hand, correlated negatively with their intake levels of fatty fish (Rho = −0.15, *p* = 0.04), total fish (Rho = −0.15, *p* = 0.04), and cow’s milk (Rho = −0.17, *p* = 0.02). 

## 4. Discussion

In the present study, we show that the fatty acid profiles of phospholipids in the cord blood samples obtained from infants at birth are linked to a modified risk of developing atopic eczema during the first year of life. More specifically, higher proportions of n-6 LCPUFAs in the infant’s phospholipids at birth were associated with an increased risk of atopic eczema at 12 months of age, whereas higher proportions of n-3 PUFAs were associated with a decreased risk. The strongest association were found for the n-6 PUFA arachidonic acid where one IQR (interquartile range) increase in the proportion of arachidonic acid in infant cord plasma was associated with an over two-fold higher risk of eczema during the first year of life, after adjustment for allergic heredity as well as after adjustment for *FLG* loss of function mutations.

Since we examined the proportions of fatty acids among plasma phospholipids, a high proportion of one type of fatty acid inevitably yields lower proportions of the other types of fatty acids. Therefore, the apparent protective effects of α-linoleic acid and DPA with regards to atopic eczema may be due to a relatively lower percentage of arachidonic acid and vice versa. When α-linolenic acid and DPA were added to a logistic regression model together with arachidonic acid these fatty acids were no longer associated with lower odds of developing atopic eczema. Furthermore, as shown in Appendix A, arachidonic acid was found to be more important than any of the n-3 fatty acids in terms of distinguishing infants with and without eczema. This indicates that arachidonic acid plays a negative role in eczema development in itself, and not only as a result of relatively lower n-3 PUFA proportions.

Fatty acids may play an important role both in atopic eczema development and disease. One association between fatty acids and atopic eczema is connected to the skin barrier function. The primary role of the skin is to prevent water loss and protect from UV light, entry of harmful compounds and organisms from the environment. The skin consists among other things of different lipids which contribute to skin barrier function, membrane structure, and cell function [25] The lipids in the extracellular lipid membranes of the outer layer of the skin, the stratum corneum, mediate the permeability barrier function [26]. Impaired skin barrier function is one important factor in atopic disease development [27]. It may, for example, increase the risk for sensitization to food allergens, according to “the dual exposure hypothesis”, which suggests that tolerance to food antigens occurs in early life through high-dose oral exposure while a low dose cutaneous exposure of food allergens through a disrupted skin barrier increases allergic sensitization [28]. *FLG* loss-of-function mutations reduce the skin barrier function and these mutations have been identified as a risk factor for different allergic diseases such as atopic eczema [29,30]. No associations were found in this study between *FLG* and atopic eczema. Further, adjusting the logistic regression models for *FLG* (or allergy within the family which reflects heredity) did not influence the association between arachidonic acid and atopic eczema, suggesting that the association between arachidonic acid and increased risk of atopic eczema is not confounded by genetic variation in skin barrier function. However, the small sample size limits our conclusions.

Except for the effect on skin permeability, fatty acids have other characteristics of importance for allergy development and disease. The association between arachidonic acid and atopic eczema development found in this study may, for example, be attributed to lipid mediators (i.e., eicosanoids), which are produced from long-chain PUFAs such as arachidonic acid [6]. For instance, prostaglandin E2 (PGE2) play important roles in several mechanisms linked to allergic sensitization and allergic inflammation. It supports the activation of dendritic cells, and suppresses the functions of macrophages, neutrophils, and the Th1-, CTL-, and NK cell-mediated type 1 immunity. In addition, it promotes vascular permeability, as well as Th2, Th17, and regulatory T-cell responses [31], while it inhibits the production of Th1-type cytokines. PGE2 is also involved in the priming of naïve T cells, resulting in the production of interleukin (IL)-4 and IL-5, as well as the promotion of immunoglobulin (Ig) class switching in naive B cells towards the production of IgE [31]. Results from previous observational studies on the effects of fatty acids in infant cord plasma on allergy development are inconsistent [32,33,34,35,36]. In accordance with our results, we have previously reported that in the FARMFLORA birth cohort, that the proportions of n-3 LCPUFAs in infant cord serum, which correlated positively with maternal fish intake during pregnancy, were negatively associated with infant allergies at both 3 and 8 years of age [32]. In addition, a Spanish cohort [33] that included 211 mother-child pairs found that both DHA and the sum of total n-3 PUFAs correlated negatively with parental-reported atopic eczema at 14 months of age. They also reported a non-significant correlation between higher levels of arachidonic acid and lower prevalence of atopic eczema. Furthermore, in a nested case-control study of 70 children, the children with allergic sensitization and atopic dermatitis before the age of 3 years were shown to have lower levels of both EPA and α-linoleic acid in their cord blood plasma [36]. In a larger cohort study that comprised 1238 infants from the ALPSAC study, no associations were found for specific fatty acids, except for the ratio of arachidonic acid to EPA in infant’s red blood cells at birth, which was positively associated with parental-reported eczema at 30 months of age [34]. In another European birth cohort, the Munich LISAplus cohort, no associations were found in 436 children between the concentrations of any n-3 or n-6 fatty acids in the infant cord serum and allergy (including eczema) at 6 and 10 years of age [35]. Hence, the results from previous epidemiological studies are inconsistent, although the majority suggests a protective effect of n-3. PUFA and further evaluation is needed.

During pregnancy, fatty acids are transported from the maternal circulation to the fetal circulation across the placenta, and for long-chain PUFAs, this is achieved through active transport mediated by transport proteins [37]. This active transport is evident when comparing the proportions of fatty acids in infant and maternal plasma, in that we found that the infant plasma contained lower proportions of linoleic acid, α-linolenic acid, EPA, and DPA but higher proportions of DGLA, arachidonic acid, adrenic acid, and DHA. These results are in accordance with our previous findings in the FARMFLORA study [38]. In the present study, we show that even though the proportions of many of the fatty acids differed between mothers and infants, all the fatty acids, with the exception of α-linolenic acid, were strongly positively correlated between infants and mothers. Therefore, the fatty acid composition of the fetus’ blood is related to the fatty acid composition of the mother’s blood, which in turn is related to the maternal intake of fat-rich food items and maternal fatty acid metabolism. Long-chain PUFAs of both the n-3 and n-6 series may be endogenously produced from short-chain fatty acid precursors, linoleic acid and α-linolenic acid, and the capacity for this conversion seems to be higher in women during pregnancy [7,8]. Fetal endogenous production of long-chain fatty acids has also been suggested, although it is not as well-studied and may be lower than maternal production [39]. When we used PLS to study how the proportions of n-3 and n-6 PUFAs in infant cord plasma and maternal plasma were related to maternal food intake, we found a positive association between n-3 PUFAs in both the infant and maternal plasma and maternal intake levels of seafood, total fish, and fatty fish, which are known sources of n-3 LCPUFAs. Moreover, a positive association between n-6 LCPUFAs and maternal intake of meat was detected.

In the present study, atopic eczema was associated with lower proportions of n-3 PUFAs and higher proportions of n-6 PUFAs, which in turn were associated with lower maternal intake of fish and higher intake of meat. These findings may suggest that maternal intake of fish during pregnancy is associated with a reduced risk of atopic eczema, and that maternal intake of meat during pregnancy is associated with an increased risk of atopic eczema during the first year of life. We have previously published data on the associations between the intake of food items and nutrients during pregnancy (and lactation) and infant allergy risk during the first year of life, among all the infants in the NICE cohort with data available for maternal food intake and allergy outcomes (N = 508). In the current study, a subset of the cohort was included based on the availability of cord plasma samples from the infants at birth and allergy outcomes (N = 206). In the previous study [19], as well as in the current study, no significant direct associations were identified between maternal fish intake and atopic eczema. This may reflect the difficulties associated with accurately measuring food intake using food frequency questionnaires [40,41,42]. Since n-3 LCPUFAs, especially EPA and DHA, are recognized biomarkers of fish intake [43,44], they may be well-suited to serve as a proxy for maternal fish intake. Thus, the inverse association observed between n-3 PUFAs and allergy in the current study may either arise from a protective effect of n-3 PUFAs *per se*. Alternatively, it could be a marker of higher maternal intake of fish, which may offer protection by means other than its content of n-3 PUFAs. The association seen between higher proportions of n-6 PUFAs and increased risk of atopic eczema is not as easily explained by maternal food intake, since arachidonic acid is found in many different food items in addition to meat. However, we found a correlation between meat intake and higher plasma arachidonic acid proportions, and this association has also been reported in previous studies [45,46]. Furthermore, the hypothesis that maternal fish intake during pregnancy protects against childhood atopic eczema is supported by several other observational studies [15,16,47,48], while the negative effects on infant allergy of maternal intake of meat are less widely reported. Still, a study on 1140 mother-infant pairs from the French EDEN cohort found that a higher maternal intake of meat during pregnancy was associated with increased risks of wheezing, allergic rhinitis, and atopic eczema in children at 3 years of age [49]. Several review articles have discussed the effects of maternal dietary intake during pregnancy on allergic outcomes in children, reaching different conclusions [6,50,51,52]. A systematic review by Kremmyda and colleagues in 2011 [53] reported on a protective association between maternal fish intake during pregnancy and allergic outcomes in the offspring based on five observational studies, while a systematic review by Netting and coworkers in 2014 [52] did not find any consistent linkages between mothers’ dietary intake and atopic outcomes in their children. In 2017, another review was published by Venter et al. [50], who concluded that current evidence regarding the effects of maternal diet in pregnancy and lactation on allergic disease outcomes in the offspring is limited [50]. Our own conclusion is that there seems to be some beneficial association between maternal fish intake during pregnancy and allergy development in offspring, and this is supported by the results of the current study, but whether or not these effects are due to a protective effect of the n-3 fatty acids *per se* or due to other components of the fish remains to be elucidated.

### Strengths and Limitations

The major strength of the present study is that atopic eczema was physician-diagnosed based on strict criteria and that all the children met with the same pediatrician specialized in allergy. A limitation of the current study is that the phospholipid fatty acids 16:0 and 18:0 could not be quantified due to contamination from the SPE columns. Therefore, as these are not included in the sum of the total phospholipids, the proportions of the different fatty acids that we show here are therefore not comparable with the levels or proportions of fatty acids reported in other studies. Furthermore, due to the observational design of our study, there is always a risk of residual confounding, and we cannot investigate causality.

## 5. Conclusions

Our results suggest that higher proportions of n-3 PUFAs and lower proportions of n-6 PUFAs in infant cord serum at birth may be associated with a reduced risk of developing atopic eczema during the first year of life. The proportions of n-3 PUFAs increase with a higher intake of fish, while the n-6 PUFAs are associated with meat consumption. Therefore, our findings imply that it might be beneficial, at least partly, to choose fish over meat during pregnancy, so as to prevent eczema in the offspring. However, it remains to be discovered whether this is due to a protective effect of the n-3 fatty acids *per se* or due to other components of the fish or if it is due to an allergy-promoting effect of arachidonic acid or meat intake.

## Figures and Tables

**Figure 1 nutrients-13-03779-f001:**
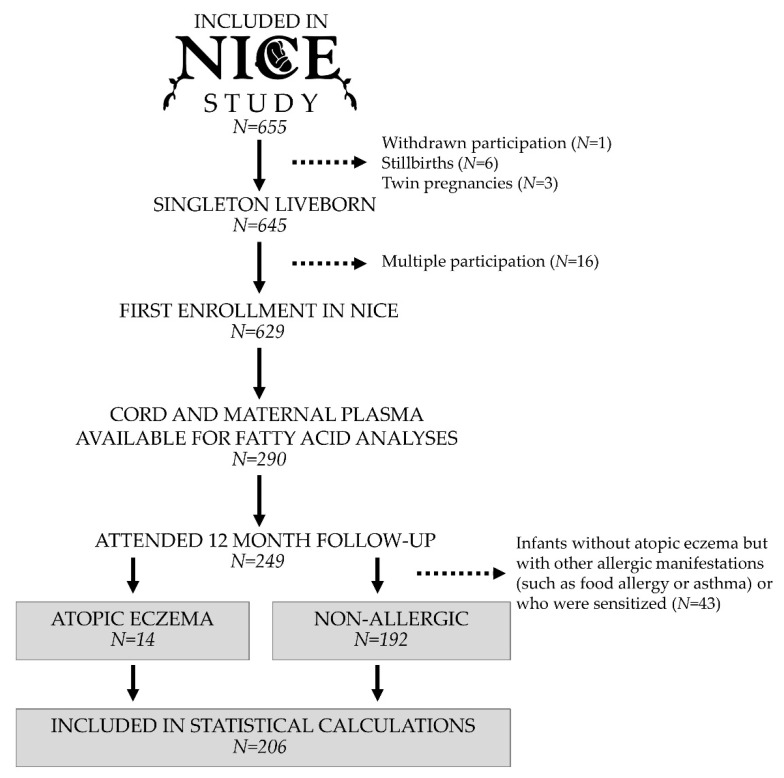
Flow chart of the study population.

**Figure 2 nutrients-13-03779-f002:**
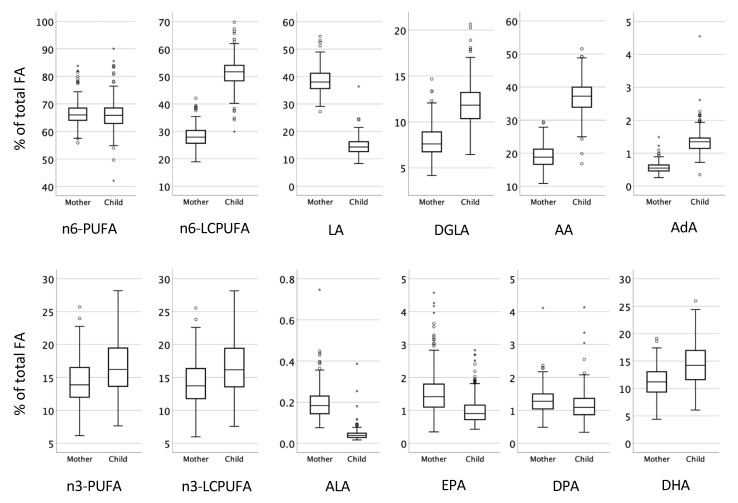
Proportions of fatty acids in the maternal and infant plasma samples at delivery. Abbreviations: PUFA, polyunsaturated fatty acids; LCPUFA, long-chain PUFA; LA, linolenic acid (18:2 n-6); DGLA, dihomo-gamma-linoleic acid (20:3 n-6); AA, arachidonic acid (20:4 n-6); AdA, Adrenic acid (22:4 n-6); ALA, α-linolenic acid (18:3 n-3); EPA, eicosapentaenoic acid (20:5 n-3); DPA, docosapentaenoic acid (22:5 n-3); DHA, docosahexaenoic acid (22:6 n-3).

**Figure 3 nutrients-13-03779-f003:**
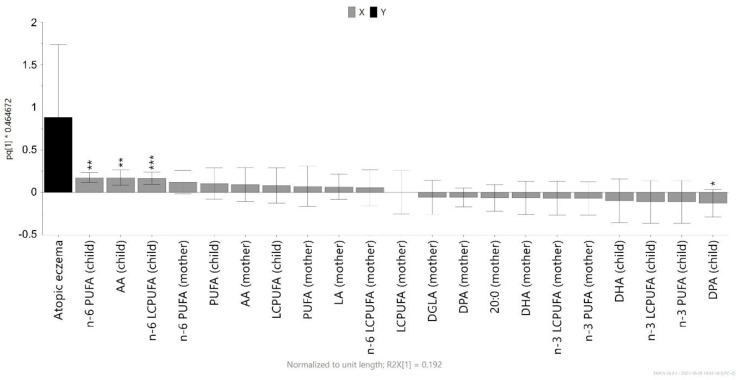
Orthogonal partial least squares (OPLS) loading column plot showing the associations between atopic eczema at 12 months of age and the proportion of selected fatty acids in maternal and infant cord plasma phospholipids. Fatty acids with a variable of importance (VIP) value >0.8 in the model that included all fatty acids (shown in Appendix A) are included here. Associations that were significant in the univariate analyses (Mann–Whitney *U*-test) are indicated with an asterisk: * *p* < 0.05, ** *p* < 0.01, *** *p* < 0.001. Abbreviations: AA, arachidonic acid (20:4 n-6); DGLA, dihomo-gamma-linoleic acid (20:3 n-6); DHA, docosahexaenoic acid (22:6 n-3); DPA, docosapentaenoic acid (22:5 n-3); LCPUFA, long-chain PUFA; PUFA, polyunsaturated fatty acids.

**Figure 4 nutrients-13-03779-f004:**
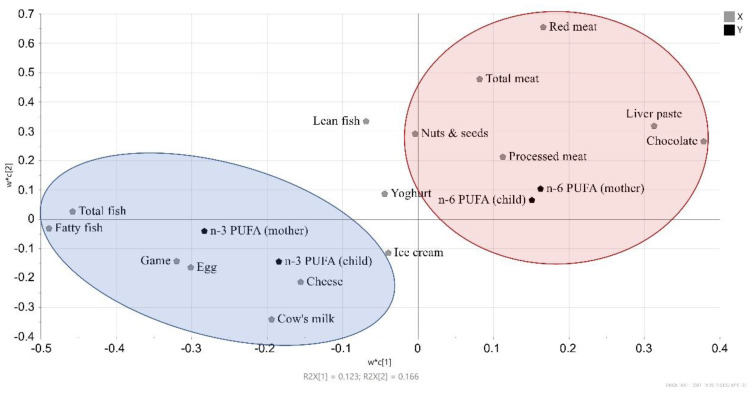
Partial least squares (PLS) loading plot showing the associations between maternal food intake during pregnancy and proportions of n-3 and n-6 polyunsaturated fatty acids (PUFAs) in the infant’s and mother’s plasma phospholipids at birth (N = 196 mothers and N = 196 children).

**Table 1 nutrients-13-03779-t001:** Characteristics of infants with atopic eczema and non-allergic infants.

	Non-Allergic (N = 192)		Atopic Eczema (N = 14)		*p*-Value
	N	(%)	N	(%)	
Male sex	82	(43)	5	(36)	0.609
Birth order					
First-born	92	(48)	8	(57)	0.505
Mother with ≥1 previous pregnancy	100	(52)	6	(43)	
Birthweight in grams					
<2500	1	(<1)	1	(7)	0.068
2500–4500	178	(93)	13	(93)	
>4500	13	(7)	0	(0)	
*FLG* null					
Heterozygous (*FLG* null)	11	(6)	0	(0)	
Homozygous Wild-Type (WT)	159	(94)	11	(100)	1.000
Breastfed					
Never	6	(3)	0	(0)	0.467
<4 months	13	(7)	2	(14)	
4–5 months	31	(17)	4	(29)	
≥6 months	130	(72)	8	(57)	
Missing	12		-		
Pet ownership (first year of life)					
Dog	62	(32)	4	(29)	1.000
Cat	47	(25)	3	(21)	1.000
Other	11	(6)	0	(0)	1.000
Allergic heredity					
Maternal	74	(39)	9	(64)	0.058
Paternal	80	(42)	9	(64)	0.099
Sibling	28	(15)	5	(71)	0.007
Any	125	(65)	13	(93)	0.033
Maternal age at delivery (years)					
<25	19	(10)	2	(14)	0.815
26–30	89	(46)	5	(36)	
31–35	56	(29)	6	(43)	
>35	28	(15)	1	(7)	
Mother’s highest education level					
Elementary school (9 years)	3	(2)	0	(0)	0.138
Senior high school (12 years)	51	(27)	7	(50)	
University/other education (>12 years)	138	(72)	7	(50)	
Early pregnancy BMI, kg/m^2^					
Underweight (<18.5)	0	(0)	0	(0)	0.910
Normal weight (18.5–24.9)	82	(56)	7	(54)	
Overweight (25–29.9)	46	(29)	4	(31)	
Obese (≥30)	28	(15)	2	(15)	
Missing	6		1		
Residential address					
Town (central part)	82	(43)	5	(36)	0.696
Town (suburb)	46	(24)	4	(29)	
Countryside	64	(33)	5	(36)	
Maternal smoking before pregnancy					
Yes	11	(6)	0	(0)	1.000
No	180	(94)	14	(100)	
Missing	1		-		
Gestational length					
Preterm	4	(2)	1	(7)	0.365
Term	167	(87)	12	(86)	
Post-term	21	(11)	1	(7)	
Birth mode					
Vaginal delivery	180	(94)	12	(86)	0.244
Cesarean section	12	(6)	2	(14)	

Differences between allergic and non-allergic children were analyzed with the chi-square test. For dichotomized variables, Fisher’s exact test or Pearson’s chi-square test was used depending on the number of expected cases in each group. For analysis of trends in categorical data, Linear-by-linear associations were used. Birth order was categorized as ‘no previous children’ (nulliparous) and ‘one or more previous children’ (multiparous). The child was defined as having allergic heredity if the mother, father, or any sibling had any diagnosis of atopic eczema, food allergy, allergic rhinoconjunctivitis, and/or asthma with treatment. Smoking was categorized as ‘yes’ or ‘no’ regardless of the number of smoked cigarettes. Gestational age (presented as days of gestation from ultrasound or, if missing, from last menstrual period) was categorized as: preterm (GW < 36+6), term (GW 37+0–41+6), and post-term (GW ≥ 42+0).

**Table 2 nutrients-13-03779-t002:** Associations between one unit increase in IQR for the proportions of different polyunsaturated fatty acids (PUFAs) in cord serum phospholipids and atopic eczema during the first year of life.

		Association with Atopic Eczema During 1st Year of Life					
		Unadjusted Model		Adjusted Model ^1^		Adjusted Model ^2^	
Fatty Acid	% in Cord Blood Phospholipids Median (IQR)	OR ^3^ (95% CI)	*p*-Value	OR ^3^ (95% CI)	*p*-Value	OR ^3^ (95% CI)	*p*-Value
n-6 fatty acids							
Linoleic acid, 18:2 n-6	14 (3.7)	1.08 (0.66–1.79)	0.754	1.01 (0.61–1.68)	0.956	0.80 (0.36–1.76)	0.578
Arachidonic acid, 20:4 n-6	37 (6.1)	2.75 (1.38–5.47)	**0.004**	2.58 (1.32–5.04)	**0.005**	2.61 (1.21–5.64)	**0.014**
n-6 LCPUFAs	52 (5.7)	2.11 (1.23–3.63)	**0.007**	1.99 (1.17–3.39)	**0.012**	1.91 (1.05–3.48)	**0.035**
n-6 PUFAs	66 (5.6)	1.79 (1.18–2.70)	**0.006**	1.67 (1.11–2.51)	**0.015**	1.52 (0.94–2.46)	0.086
n-3 fatty acids							
α-linolenic acid, 18:3 n-3	0.039 (0.021)	0.33 (0.12–0.93)	**0.036**	0.34 (0.12–0.94)	**0.038**	0.39 (0.13–1.19)	0.099
EPA, 20:5 n-3	0.92 (0.48)	0.80 (0.41–1.53)	0.494	0.74 (0.38–1.44)	0.371	0.81 (0.39–1.70)	0.583
DPA, 22:5 n-3	1.1 (0.52)	0.41 (0.17–1.01)	0.052	0.40 (0.16–1.00)	**0.049**	0.56 (0.22–1.40)	0.215
DHA, 22:6 n-3	14 (5.5)	0.53 (0.22–1.29)	0.160	0.53 (0.22–1.30)	0.165	0.83 (0.32–2.16)	0.699
n-3 LCPUFAs	16 (5.9)	0.49 (0.20–1.19)	0.114	0.48 (0.20–1.18)	0.109	0.76 (0.30–1.95)	0.571
n-3 PUFAs	16 (5.9)	0.49 (0.20–1.18)	0.111	0.48 (0.20–1.17)	0.107	0.76 (0.29–1.95)	0.563

Atopic eczema was diagnosed at 12 months of age through clinical examination by an experienced pediatric allergologist. The proportions of various PUFAs were measured in the phospholipid fractions of cord plasma samples collected at delivery and related by multiple logistic regression to the occurrence of infant atopic eczema during the first year of life. ^1^ Adjusted for any allergic disease within the family (sibling or parent). ^2^ Adjusted for *FLG* loss-of-function mutations. ^3^ OR per interquartile range (IQR) of fatty acid proportions. Both the unadjusted and the model adjusted for allergy within the family included 14 infants with eczema and 192 non-allergic infants, the model adjusted for *FLG* mutations included 11 infants with atopic eczema and 170 non-allergic individuals. Significant *p*-values are indicated in bold. Abbreviations: CI, confidence interval; OR, odds ratio; PUFA, polyunsaturated fatty acids; LCPUFA, long-chain PUFA; EPA, eicosapentaenoic acid; DPA, docosapentaenoic acid; DHA, docosahexaenoic acid.

## Data Availability

The food frequency questionnaires are not publicly available due to proprietary rights. The raw data used in this study are not publicly available because they relate to information that could compromise research participant privacy or consent. Explicit consent to deposit raw data was not obtained from the participants. Therefore, the data can only be made public if a new consent is filled in by the participants together with a new ethical permit being obtained.

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
