# Peer review of "Proportions of Polyunsaturated Fatty Acids in Umbilical Cord Blood at Birth Are Related to Atopic Eczema Development in the First Year of Life"

_nutrients, 2021, doi:10.3390/nu13113779_

Round 1

Reviewer 1 Report

What more adjusted in addition to allergy history among factors that can affect atopic dermatitis occurrence? Did other factors have not been adjusted because there is no difference between the two groups?

Is there anything about nutrients (e.g lactobacillus) during the survey?

Author Response

What more adjusted in addition to allergy history among factors that can affect atopic dermatitis occurrence? Did other factors have not been adjusted because there is no difference between the two groups?

Is there anything about nutrients (e.g lactobacillus) during the survey?

Authors comment:

Dear reviewer,

thanks for taking the time to review or manuscript and thanks for your helpful comments.

We did adjust for allergy within the family since this was the only characteristic variable that differed between allergic and non-allergic individuals.

After a comment from another reviewer, we have also added filaggrin loss-of-function mutations to our manuscript and added FLG as a confounder in a separate adjusted logistic regression model and discussed the results from this in the paper.

We do have data on nutrients and on intake of lactobacillus containing food products and supplements. But this is not within the scope of the current manuscript and is not included here. We have reported on maternal dietary intake in a previous publication (Stravik, M.; Barman, M.; Hesselmar, B.; Sandin, A.; Wold, A.E.; Sandberg, A.S. Maternal Intake of Cow's Milk during Lactation Is Associated with Lower Prevalence of Food Allergy in Offspring. Nutrients 2020, 12, 1-19.) and we are planning to publish a paper about the infants’ dietary intakes later on.

Reviewer 2 Report

The question addressed in the manuscript was to determine whether maternal intake of fatty acids during pregnancy was correlated with the proportions of fatty acids among plasma phospholipids of the mother and the infant at birth and correlated to the development of atopic eczema during the first year of age.

I have no major concern. The manuscript is very well written, the methods are well described, require patient outcomes are mentions, statistical analysis are accurate.

- Was the transepidermal water loss (TEWL) measured during the clinical examination at the time of eczema diagnosis ?

- Was the filaggrin status of the child known ?

Indeed, PUFAs are known to have effect on skin barrier and may be of great interest for inflammatory skin diseases acting on the epithelial barrier functions (permeability) and not only on the immune system. Effects of PUFAs on epithelial barrier should be discussed.   

Author Response

Reviewer 1:

The question addressed in the manuscript was to determine whether maternal intake of fatty acids during pregnancy was correlated with the proportions of fatty acids among plasma phospholipids of the mother and the infant at birth and correlated to the development of atopic eczema during the first year of age.

I have no major concern. The manuscript is very well written, the methods are well described, require patient outcomes are mentions, statistical analysis are accurate.

- Was the transepidermal water loss (TEWL) measured during the clinical examination at the time of eczema diagnosis ?

-Was the filaggrin status of the child known ?

Indeed, PUFAs are known to have effect on skin barrier and may be of great interest for inflammatory skin diseases acting on the epithelial barrier functions (permeability) and not only on the immune system. Effects of PUFAs on epithelial barrier should be discussed.

Authors comment:

Dear reviewer,

thanks for taking the time to review our manuscript and thanks for your helpful comments.

Regarding TEWL, we have not measured this.

Filaggrin status is however known on the children, and we have added data on filaggrin (FLG) loss-of-function mutations in the manuscript. We have also added FLG as a confounder in a separate adjusted logistic regression model and discussed the results from this in the paper.

We agree that fatty acids may affect skin barrier and we have added a section about this both in the introduction and in the discussion. Line 41-45 in the introduction and line 402-421 in the discussion (clean version line numbers).